# Multistage coupling water-enabled electric generator with customizable energy output

Puying Li[1], Yajie Hu [ID][1], Wenya He [ID][1], Bing Lu[1], Haiyan Wang[1], Huhu Cheng [ID][1] ✉ & Liangti Qu [ID][1] ✉

Constant water circulation between land, ocean and atmosphere contains great and sustainable energy, which has been successfully employed to generate electricity by the burgeoning water-enabled electric generator. However, water in various forms (e.g. liquid, moisture) is inevitably discharged after one-time use in current single-stage water-enabled electric generators, resulting in the huge waste of inherent energy within water circulation. Herein, a multistage coupling water-enabled electric generator is proposed, which utilizes the internal liquid flow and subsequently generated moisture to produce electricity synchronously, achieving a maximum output power density of ~92 mW m$^{-2}$ (~11 W m$^{-3}$). Furthermore, a distributary design for internal water in different forms enables the integration of water-flow-enabled and moisture-diffusion-enabled electricity generation layers into mc-WEG by a "flexible building blocks" strategy. Through a three-stage adjustment process encompassing size control, space optimization, and large-scale integration, the multistage coupling water-enabled electric generator realizes the customized electricity output for diverse electronics. Twenty-two units connected in series can deliver ~10 V and ~280 μA, which can directly lighten a table lamp for 30 min without aforehand capacitor charging. In addition, multistage coupling water-enabled electric generators exhibit excellent flexibility and environmental adaptability, providing a way for the development of water-enabled electric generators.

Water constantly circulates between land, ocean and atmosphere in the form of gas, liquid and solid on the earth, which contains great and sustainable energy to be exploited[1–6]. Recently, the emerging water-enabled electric generator (WEG) and hydrovoltaic technology have been developed with the interaction between functional materials and different forms' water[2,6–10]. For example, moisture-enabled electric generators (MEGs) based on the ion diffusion generated in graphene oxide convert the variation in chemical potential of water into electricity[6,11–16]. Streaming current based on the electric double layer at the interface between the solution and the nanochannels can be generated when the liquid flows inside nanostructured materials caused by water evaporation or external pressure[17–24]. Electricity is also generated on the surface of multi-wall carbon nanotubes and the like by the movements of water droplets[25–27]. However, water in different forms (e.g. liquid, moisture) will be inevitably discharged after disposable employment for electricity generation by currently single-stage WEGs above mentioned, resulting in the heavy waste of inherent energy within water circulation. In nature, even in the process of water absorption by arid soil with low water potential[28,29], not only liquid flow but also procreant moisture will diffuse from wet regions into dry regions (Fig. 1a), which involves multistep water utilization and takes advantage of water resources adequately.

[1]Laboratory of Flexible Electronics Technology, Key Laboratory of Organic Optoelectronics & Molecular Engineering, Ministry of Education, Department of Chemistry, State Key Laboratory of Tribology in Advanced Equipment (SKLT), Tsinghua University, Beijing 100084, P. R. China. ✉e-mail: huhucheng@tsinghua.edu.cn; lqu@mail.tsinghua.edu.cn

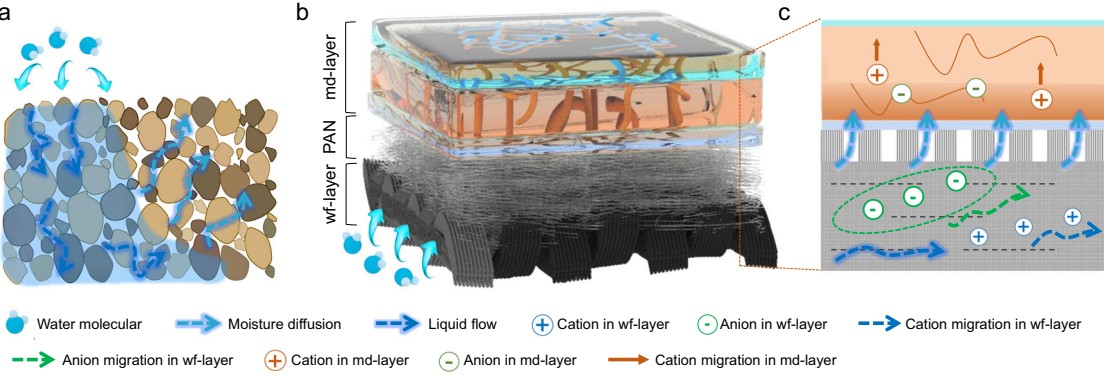

**Fig. 1 | Schematic diagram of the mc-WEG. a** Water absorption process of arid soil. The liquid flows along the tiny channels in the soil from wet region toward the dry region. At the same time, in areas where the liquid cannot flow through, water will be transferred in the form of moisture diffusion. **b** Schematic diagram of the structure of mc-WEG. **c** The water transportation track in mc-WEG. Bottom wf-layer utilize the flow of liquid to generate electricity, and the top md-layer is based on the moisture induced ion migration for electricity generation. A porous polyacrylonitrile membrane is placed between wf-layer and md-layer for water diversion.

In this work, a multistage coupling water-enabled electric generator (mc-WEG) is developed based on the multifunctional layers' construction and water diversion design, which can spontaneously make sufficient utilization of the inner liquid flow and concomitant moisture diffusion to produce electricity synchronously (Fig. 1b). As indicated in Fig. 1b, mc-WEG absorbs water from the air to form the liquid at one side, following electricity generated when the liquid directionally flows through the negatively-charged surfaces of fabrics in the bottom layer (named as water-flow-enabled electricity generation layer, wf-layer) (Fig. 1c). Simultaneously, the evaporated moisture from the bottom layer will diffuse into the top layer (named as moisture-diffusion-enabled electricity generation layer, md-layer), inducing ionic concentration gradients and the following directional migration of ions for further electricity production (Fig. 1c). By virtue of adequately electricity generation through the coordination of internal liquid flow and moisture diffusion, this mc-WEG exceeds the previous WEGs' limitation on utilization of single water form, achieving a maximum output power density of ~91.77 mW m$^{-2}$ (~10.92 W m$^{-3}$). In addition, wf-layer and md-layer can be assembled arbitrarily as "flexible building blocks" to construct specific mc-WEGs for various customized demands. Based on the three-stage adjustment strategy of size control, space optimization as well as integration design, variously customized mc-WEGs realize the required electricity output for different electronics of voltage-driven atomized glass and current-driven lamps. For example, twenty-two mc-WEG units connected in series can achieve ~10.32 V and ~280 μA output, driving a table lamp to continue working more than 30 min without charging the capacitor in advance. mc-WEG also endows good flexibility and environmental adaptability, which can maintain stable electrical output after being folded and exposed in natural environments. This work provides a way for the development of WEG energy harvesting system.

## Results

### Structure and internal water transmission path of mc-WEG

mc-WEG is composed of the CaCl$_2$ asymmetrically loaded carbon fabric (Fig. 2a) as wf-layer for liquid flow-enabled electricity generation, the hydrophilic polyacrylonitrile (PAN) membrane (Fig. 2b) as the diversion layer for moisture permeation from wf-layer to md-layer, and the polyelectrolyte membrane (Fig. 2c) as md-layer for moisture enabled electricity generation. Briefly, the carbon fabric was prepared by soaking a cotton fabric (~200 μm thickness, Supplementary Fig. 1a, b) in the dispersion liquid of carbon black nanoparticles (Ketjen black, Supplementary Fig. 1c, d) with abundant carboxyl groups and hydroxyl groups. Then, part of the carbon fabric was immersed in CaCl$_2$ aqueous solution (25 wt%) and dried at 80 °C to obtain the asymmetric CaCl$_2$

load (Supplementary Fig. 2). Au electrodes are connected at the ends of CaCl$_2$ loaded region and unloaded region of carbon fabric for electrical generation output (Supplementary Fig. 3). As shown in SEM images in Fig. 2a, there is no obvious difference between CaCl$_2$ loaded part and the rest part of carbon fabric in appearance, where the grain of the fabric is clearly visible. Energy dispersive spectroscopy (EDS) mapping results exhibit the significant Ca and Cl elements signals at CaCl$_2$ loaded part of carbon fabric, indicating the asymmetric CaCl$_2$ load in wf-layer. The hydrophilic PAN membrane was directly spun on an Au mesh substrate through electrospinning technique (Fig. 2b) to form the porous membrane for moisture permeation. It can be seen from Fig. 2b that the electrospun PAN nanowires are well distributed on Au mesh, exhibiting plenty of macro-channels. The hydrophilic PAN membrane on Au mesh is also employed as the bottom electrode of md-layer. md-layer was H$_2$SO$_4$-doped polystyrene sulfonic acid membrane (denoted as H-PSS membrane) sandwiched between polyvinyl alcohol (PVA) membranes containing different LiCl contents (denoted as PVA-LiCl($c$) membranes, where $c$ is the concentration of LiCl). The Cl element mapping in Fig. 2c reveals the uniform distribution of LiCl in PVA-LiCl($c$) membranes. The PVA-LiCl($c$) membranes and H-PSS membrane were prepared by a simple casting method respectively, which then directly attached to each other layer by layer (Supplementary Fig. 4) and placed between the bottom Au mesh electrode and top Au foil electrode for generation output. Finally, the wf-layer, PAN membrane and md-layer are assembled by stacking them in sequence as shown in Supplementary Fig. 3, which then being encapsulated by polyimide tape to fabricate the final mc-WEG.

When mc-WEG is exposed in moist environment, the CaCl$_2$ loaded region in wf-layer preferentially absorbs water (Fig. 2d and stage I in Supplementary Fig. 5). Subsequently, the liquid will be formed at CaCl$_2$ loaded region and spontaneously flows into the unloaded region along the carbon fabric, which then induces electricity generation caused by the diffusion imbalance of anions and cations on the surfaces because cations in the formed liquid will be preferentially adsorbed in Stern layer on wf-layer (stage II in Supplementary Fig. 5)[25,26]. Meanwhile, water in wf-layer will continuously evaporate and diffuse through porous PAN membrane into md-layer (Fig. 2d). The evaporated moisture absorbed by the md-layer whereafter motivated the formation of dissociated ion concentration gradients and the subsequently directional migration for electricity generation. In consequence, mc-WEG can employ the multistage water-flow-enabled electricity generation and moisture-diffusion-enabled electricity generation synchronously in one device.

To verify the efficient water transmission from wf-layer through PAN membrane into the md-layer of mc-WEG, the flow of liquid on the

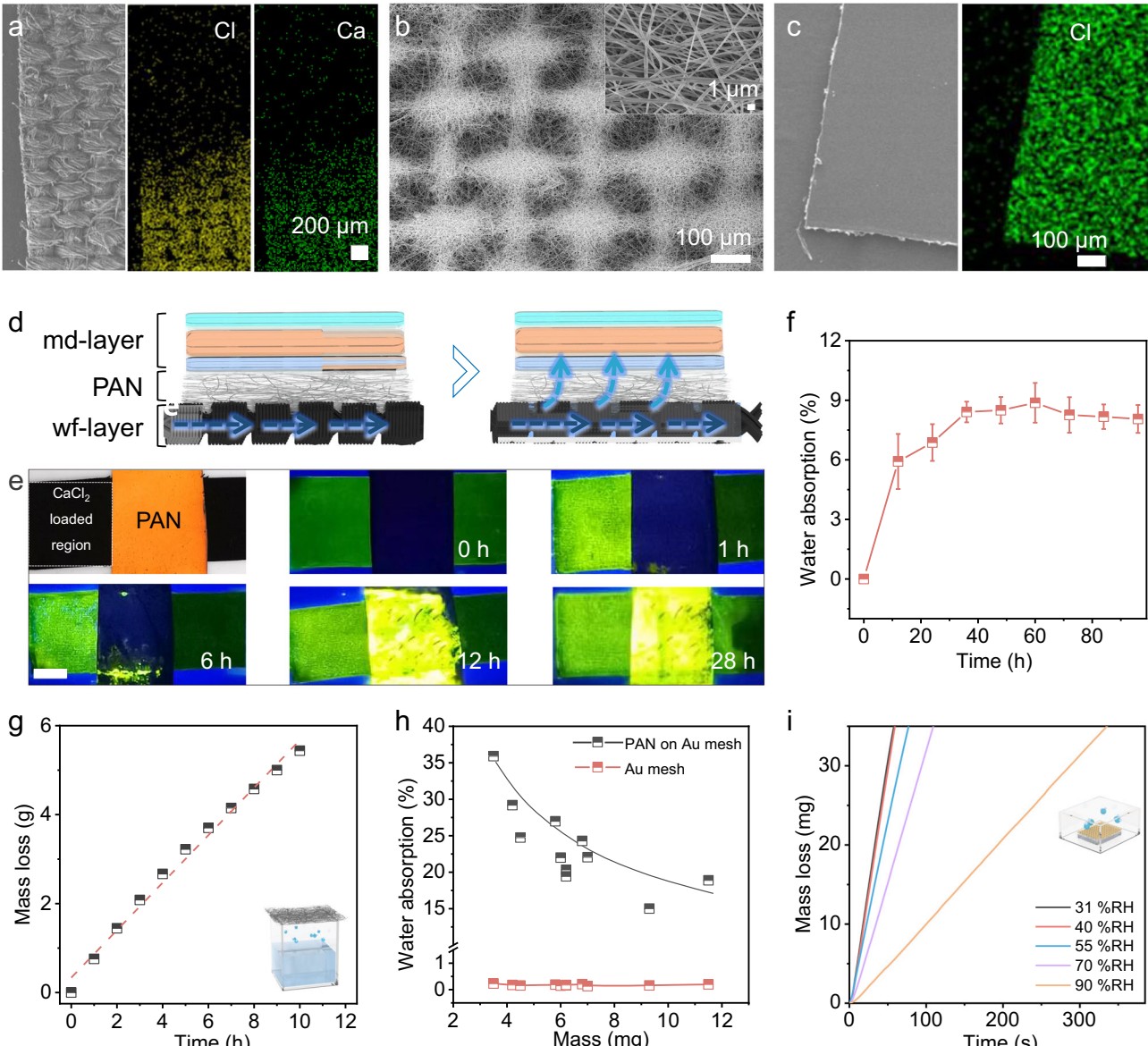

**Fig. 2 | Structure and working principle of mc-WEG. a** SEM image of wf-layer and the element mapping images of Ca and Cl. **b** SEM image of PAN membrane on the porous Au mesh substrate. **c** SEM image of PVA-LiCl (0.5 M) membrane and the element mapping image of Cl. **d** Schematic diagram of multistage water transport process in mc-WEG. Navy blue arrows represent the direction of liquid flow, and the blue arrows represent the direction of moisture diffusion. **e** Photographs of water transport on wf-layer with PAN membrane before and after being exposed at 90% relative humidity (RH) under ultraviolet light. Scale bar: 1 cm. Green regions are the water infiltrated part. **f** Water absorption of PAN membrane on wf-layer ($n$ = 3, error bars: standard deviation). **g** Moisture permeability test of PAN membrane. Inset displays the test diagram. **h** Water absorption of PAN membrane on Au mesh substrate and Au mesh only. **i** Dehumidification curves of PAN membrane (loaded on porous Au electrode) saturated with 10.08 mg ml$^{-1}$ CaCl$_2$ aqueous solution under different RH (25 °C). Inset displays the test diagram. Source data are provided as a Source Data file.

wf-layer was evaluated under ultraviolet light by pre-impregnating the wf-layer with fluorescein sodium firstly[30]. As exhibited in Supplementary Fig. 6, the CaCl$_2$ loaded region preferentially absorbs water from air to form the liquid at the beginning, manifesting strong fluorescence at CaCl$_2$ loaded region. Then liquid flows from the CaCl$_2$ loaded region into the CaCl$_2$ unloaded region along the carbon fabric of wf-layer within 28 h, demonstrated by the similar fluorescence of the whole wf-layer. While with PAN membrane on the wf-layer, PAN membrane expresses distinct fluorescence after 12 h (Fig. 2e), indicating that liquid can be absorbed by PAN membrane at the same time with the liquid flow on the wf-layer. The water absorption ($\frac{\Delta m}{m_0}$, where $\Delta m$ is the mass change of the PAN membrane, m$_0$ is the initial mass of the PAN membrane) of PAN membrane on wf-layer (Fig. 2f) gradually increases and the water absorption basically maintains at 8.37% after 40 h. It is

reasonably concluded that water will pass through the porous PAN membrane in the form of moisture diffusion in the early stage (<12 h) (Supplementary Fig. 5, stage II) and then the PAN membrane begins to act as moisture reservoir for md-layer (Supplementary Fig. 5, stage III).

The moisture transmission rate of PAN membrane (~60 μm) is 1284.98 g m$^{-2}$ h$^{-1}$ (Fig. 2g) owing to the porous structure, exhibiting favorable moisture permeability[31]. Meanwhile, the water absorption (~8.37%) of PAN membrane on wf-layer is not saturated and below the saturated water absorption of PAN membrane in the pure water (~35.91%) (Fig. 2h), which indicates that the water diversion from wf-layer to md-layer is unrestricted and would not be determined by the saturated water absorption capacity of PAN membrane. Ulteriorly, the amount of water reaching the md-layer from water saturated PAN membrane could be related to the water dehydrated capacity of PAN

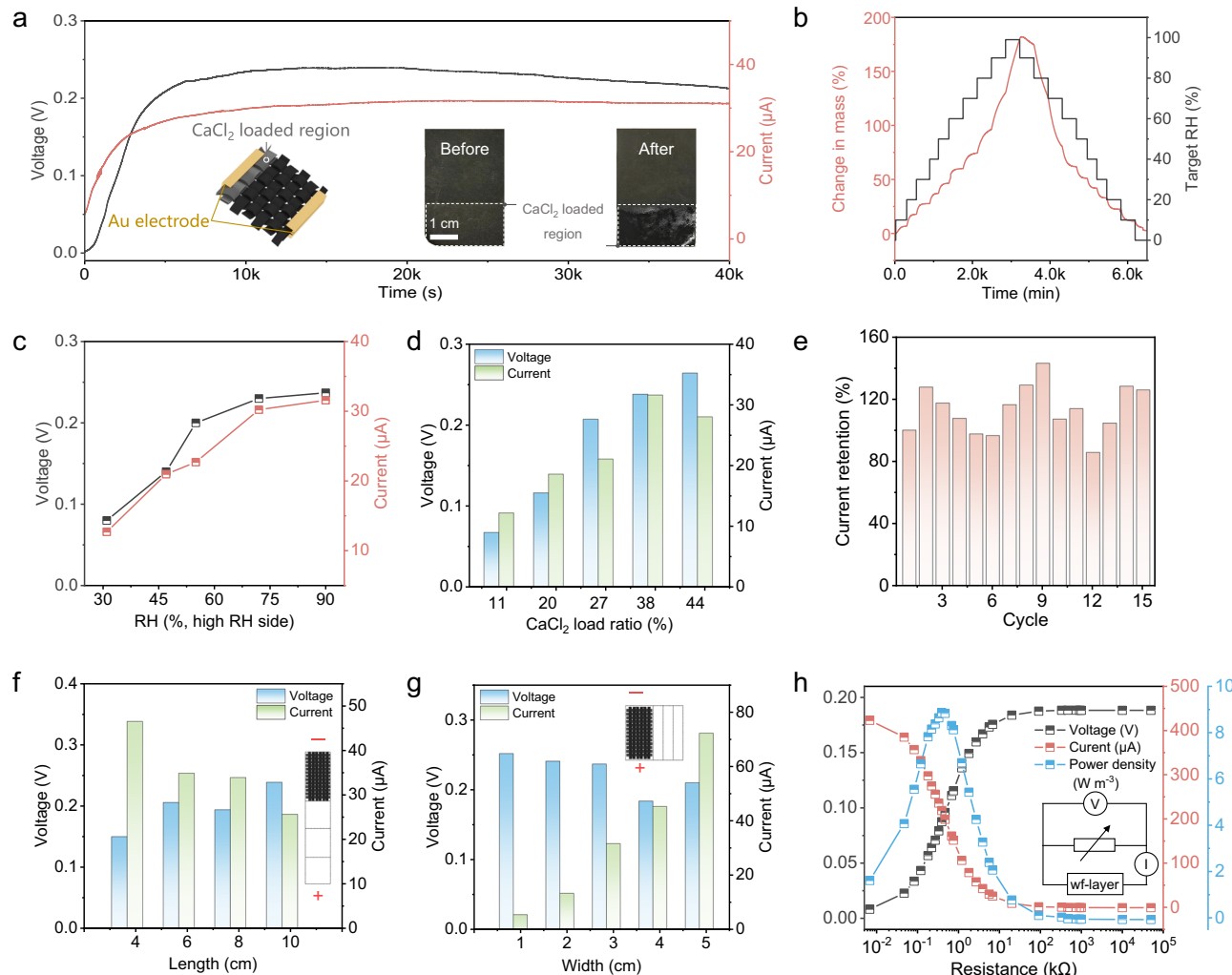

**Fig. 3 | Structure and electric generation of wf-layer. a** The voltage and current output of wf-layer (3 × 6 cm, width × length). The output can sustain 40,000 s under asymmetrical humidity (90% RH and 25% RH, the same below). Left inset exhibits the structure diagram of the wf-layer. Right inset shows the asymmetrical water absorption of wf-layer before and after being exposed in asymmetrical humidity. **b** Moisture absorption and desorption kinetics for $CaCl_2$ loaded region of wf-layer at 25 °C. **c** Voltage and current output of wf-layer (3 × 6 cm) in response to variation in RH in high RH side (25% RH for low RH side). **d** Bar graph of electricity performance of wf-layer (3 × 6 cm) with different $CaCl_2$ load ratio. **e** Current retention of wf-layer (3 × 6 cm) for different circulations. The wf-layer was dried by being placed at 80 °C for 1 h, and the output of the dried wf-layer can be restored by absorbing water again. **f, g** Voltage and current output in response to variation in length with the fixed width of 3 cm (**f**) and variation in width with the fixed length of 6 cm (**g**) under asymmetrical humidity. **h** Voltage, current and volumetric power density of wf-layer (18 × 6 cm, crimped state) with different electric resistance. Inset displays the schematic of circuit. Source data are provided as a Source Data file.

membrane. The water dehydrated capacity of the PAN membrane is about 179.91 g m⁻² h⁻¹ at 31% relative humidity (RH), which still preserves well dehydrated ability (31.3 g m⁻² h⁻¹) even at 90% RH (Fig. 2i), further confirming the excellent water diversion of PAN membrane. Hence, water transmission from wf-layer to md-layer in our mc-WEG is worthy of expectation no matter in the form of moisture penetration through PAN membrane in the early stage or water reservoir in the later stage.

**Electricity generation performance of wf-layer**

The electricity generation of wf-layer in mc-WEG based on the liquid flow on the $CaCl_2$ asymmetrically loaded carbon fabric was then studied (Fig. 3a). As $CaCl_2$ on wf-layer absorbs water from the environment, the liquid ($CaCl_2$ solution) in $CaCl_2$ loaded region flow into the $CaCl_2$ unloaded region of wf-layer and the electrical double layer will be introduced at the carbon fabric surface (Supplementary Fig. 7a). Carboxyl groups and hydroxyl groups of carbon black nanoparticles on the carbon fabric surface will produce immobilized negatively-charged $-COO^-$ and $-O^-$ groups (Supplementary Fig. 8 and 9). Cations

($Ca^{2+}$, $H^+$) in the formed liquid will be adsorbed by above negatively-charged groups, leading to the formation of Stern layer. The anions ($Cl^-$) in the diffusion layer were retarded in migration during the flow of liquid from the $CaCl_2$ loaded region to the unloaded region, resulting in an imbalance of anion and cation charges (Supplementary Fig. 7a)[25,32], giving rise to the potential difference and electricity generation[25].

The $CaCl_2$ unloaded region of wf-layer was placed in the low humidity (~25% RH) to simulate the condition in mc-WEG (Supplementary Fig. 7b). As expected, the wf-layer (3 × 6 cm, width × length) can generate ~0.24 V in open-circuit voltage ($V_{OC}$) and ~31 μA in short-circuit current ($I_{SC}$) (Fig. 3a) for a long-periods (>10,000 s) when the $CaCl_2$ loaded region of wf-layer in ~90% RH environment. The current-voltage curves further verified the $V_{OC}$ and $I_{SC}$ of wf-layer (Supplementary Fig. 10). Meanwhile, water sorption kinetics and isotherm profiles display that the $CaCl_2$ loaded region of wf-layer is able to absorb up to 182% water (25 °C) (Fig. 3b), and the liquid at the $CaCl_2$ loaded region of wf-layer can be clearly observed visually (right inset in Fig. 3a), confirming the super water absorption capacity and the

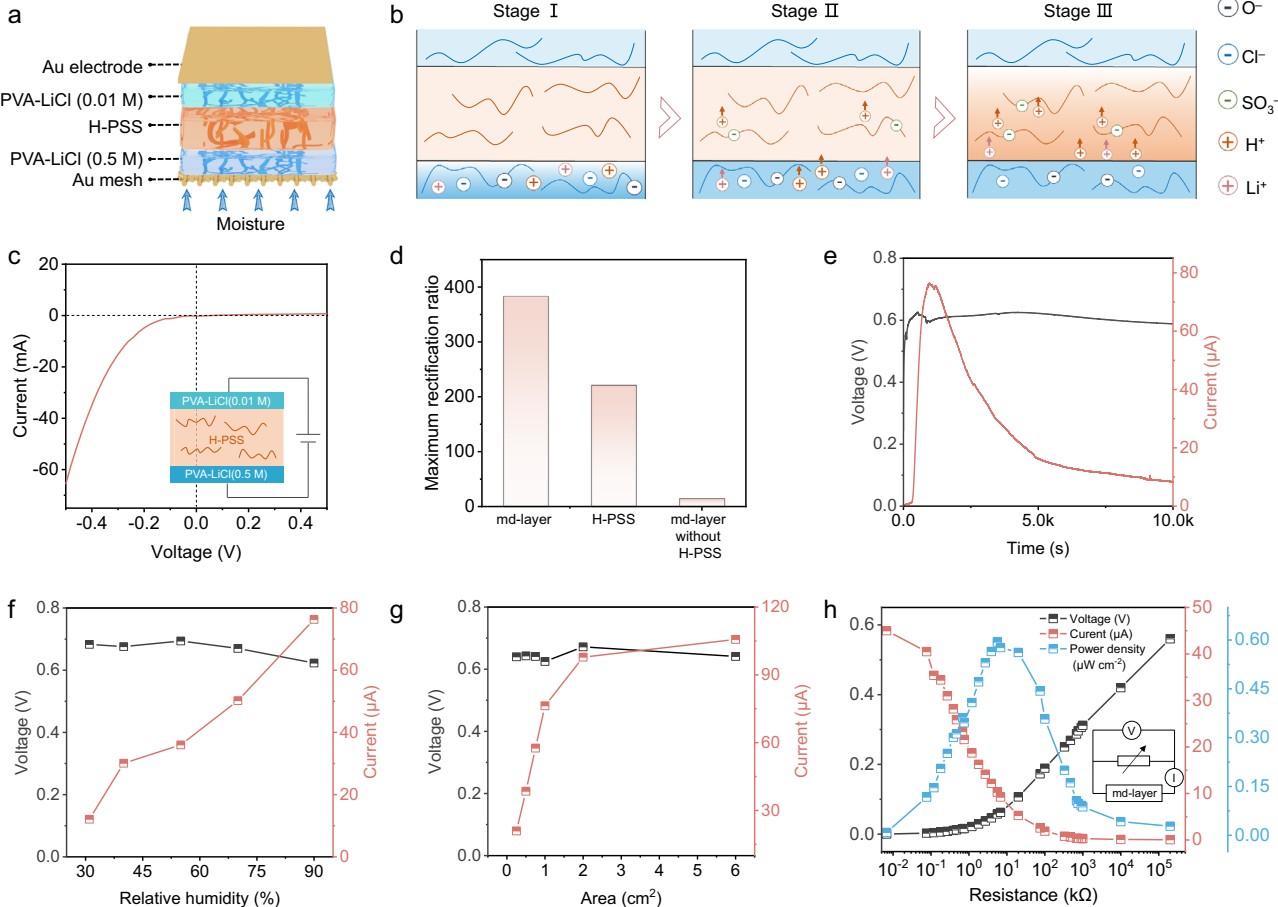

**Fig. 4 | Electric generation, working mechanism and the regulation of electricity-generating performance of md-layer. a** Scheme of the md-layer structure. The md-layer is composed of electricity-generating material (PVA-LiCl(0.5 M) membrane, H-PSS membrane and PVA-LiCl(0.01 M) membrane from bottom to top) and asymmetric Au electrodes. **b** Working mechanism of moisture-enabled electric generation in md-layer. **c** Current-voltage curve of md-layer after absorbing moisture for 1,330 s. Inset displays the schematic of circuit. **d** The maximum rectification ratio of md-layer, H-PSS membrane and md-layer without H-PSS. **e** Voltage and current output of md-layer ($1 \times 1$ cm). The electrical-generated performance can sustain 10,000 s under 90% RH 25 °C. **f** Voltage and current output of md-layer in response to variation in RH. **g** Voltage and the current of md-layer with different area. **h** Voltage, current and area power density of md-layer ($0.25 \times 0.25$ cm) with different electric resistance. Inset displays the schematic of circuit. Source data are provided as a Source Data file.

subsequent formation of liquid for electricity generation. The output performance improves from 0.08 V to 0.24 V in $V_{OC}$ and 12.70 μA to 31.58 μA in $I_{SC}$ with the humidity increment (from 31% RH to 90% RH) at the CaCl$_2$ loaded region of wf-layer (Fig. 3c), owing to the higher water absorption (Fig. 3b) resulted in the enhancement of liquid formation.

In comparison, no obvious output could be observed when the CaCl$_2$ was loaded in the middle of the carbon fabric (Supplementary Fig. 11), because the diffusion of liquid on two directions cancelled each other out. Meanwhile, no voltage and current was generated when wf-layer was put at dry condition (<15% RH) (Supplementary Fig. 12), and the pure cotton fabric was put under the test condition of wf-layer (Supplementary Fig. 13), further confirming the power generation is induced by the asymmetric CaCl$_2$ loading design and the following water absorption as well as flow on wf-layer. Besides, the moisture escape ability at CaCl$_2$ unloaded region will affect the liquid flow on wf-layer. As indicated in Supplementary Fig. 14, the lower humidity (25% RH) at CaCl$_2$ unloaded region shows the higher $V_{OC}$ of -0.24 V and $I_{SC}$ of -31 μA than that in 90% RH condition (-0.19 V, -20.44 μA), which could contribute to the rapid water escape and the subsequently enhanced liquid flow on wf-layer resulted by the low humidity. Predictably, the excellent hydroscopicity and strong water absorption ability of md-layer in mc-WEG will bring about the enhanced water escape at CaCl$_2$ unloaded region of wf-layer, therefore enhancing the liquid flow and electricity generation. It is worth noting

that, the inert Au electrodes are employed for electrical generation to avoid the corrosion (Supplementary Fig. 15)[33-35].

The load ratio of CaCl$_2$ which is crucial for the water absorption was analyzed further. As shown in Fig. 3d, $V_{OC}$ of wf-layer gradually increased from 0.07 V to 0.26 V when the CaCl$_2$ load ratio changes from 11 wt% to 44 wt%, which can be attributed to the more water absorption in higher CaCl$_2$ loaded wf-layer. The CaCl$_2$ loaded length will also influence the electrical output performance (Supplementary Fig. 16). The electric resistance of wf-layer is significant for the electrical generation. $V_{OC}$ diminished while $I_{SC}$ enhanced with the resistance dropped from 71.37 kΩ to 3.06 kΩ when rising the load of carbon black (Supplementary Fig. 17). Besides, wf-layer can be exposed at humidity environment for electricity generation again after being dried at 80 °C for water desorption, and the $I_{SC}$ maintained admirably after 15 times water absorption and desorption cycles (Fig. 3e).

In addition, the highly and easily regulated size as well as geometrical morphology of wf-layer will provide more potential for the performance adjustment and final applications of mc-WEG. With the length of wf-layer (3 cm in width) changed from 4 cm to 10 cm, the $V_{OC}$ is enhanced and the current decreased, which could be induced by the rising electric resistance (Fig. 3f). The increase in width of wf-layer (6 cm length) from 1 cm to 5 cm will give rise to the increment of larger amount of ions diffusion, thus resulting in the increase of $I_{SC}$ from 5.47 μA to 72.30 μA (Fig. 3g). Even for the wf-layer with same size, the

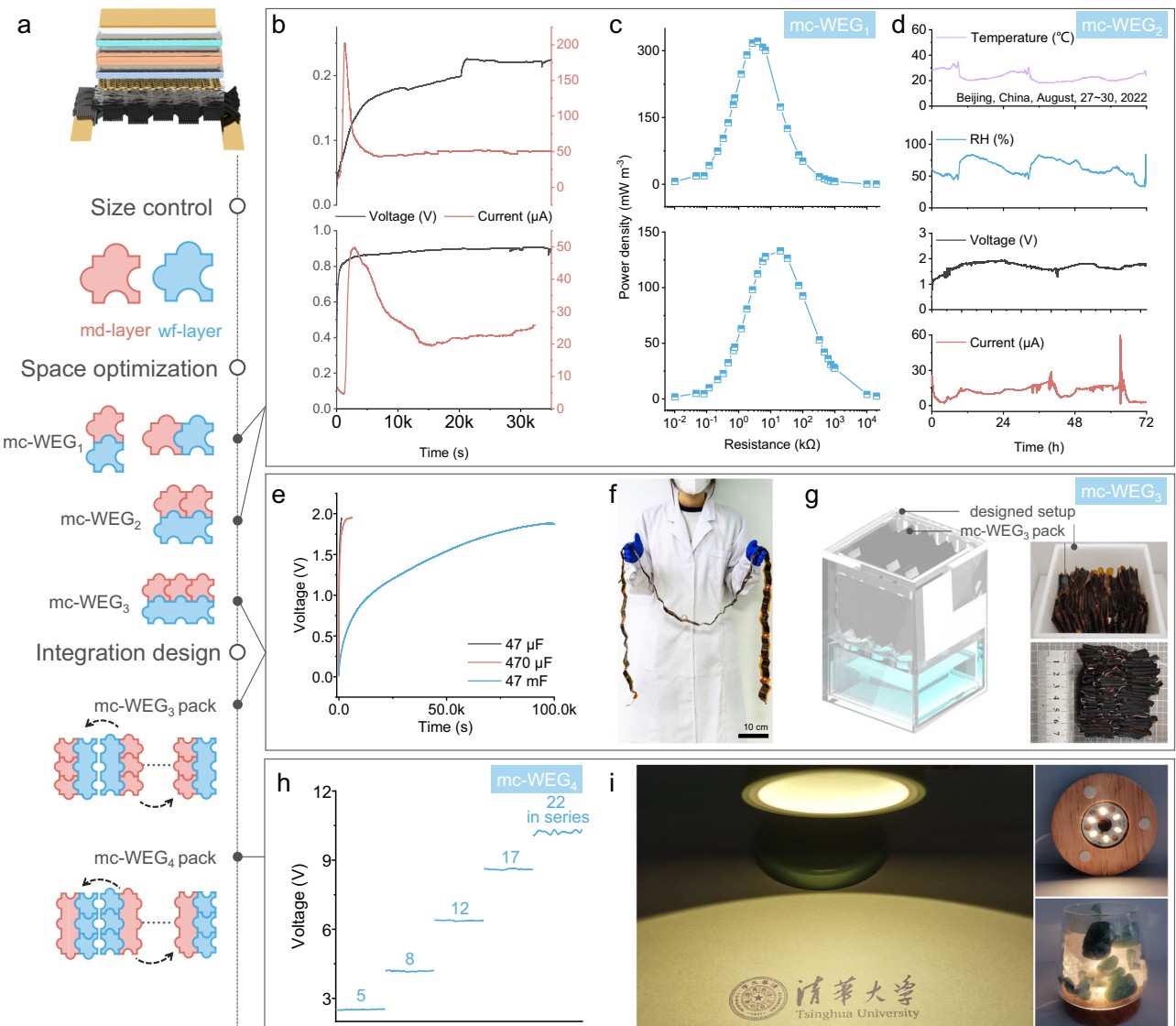

**Fig. 5 | Customized output and scalable integration of mc-WEG. a** Schematic diagram of the structure of mc-WEG and the "flexible building block" approach through size control, space optimization, as well as integration design. **b, c** Open-circuit voltage, short-circuit current (**b**) and volumetric power density with different electric resistance (**c**) of mc-WEG$_1$ in parallel connection (upper plot) and series connection (lower plot). **d** Variation of temperature, RH, as well as voltage and current generated by mc-WEG$_2$ during outdoor testing (Beijing in China: East longitude ≈116° and North latitude ≈ 40°; Time on August 27th-30th, 2022). **e** Voltage-time curves of commercial capacitors with various capacitance (0.47, 47

and 470 mF) charged by three mc-WEG$_3$ units connected in parallel. **f** Digital photo of the mc-WEG$_3$ pack before being folded (twenty-four mc-WEG$_3$ connected in series). **g** Schematic diagram of the designed auto-switchable adsorption and desorption generating setup (left diagram). Digital photos of the folded mc-WEG$_3$ pack (right). **h** The relationship between open-circuit voltage and number of mc-WEG$_4$ connected in series. **i** Digital photos of working table lamp (left) as well as LED strip (right) powered by mc-WEG$_4$ pack. Source data are provided as a Source Data file.

electrical generating performance can be modulated by the folding and crimped design (Supplementary Fig. 18). For example, the $I_{SC}$ of wf-layer (9 × 6 cm) increased from 86.95 μA to 260.62 μA when the morphology of wf-layer changed from tiled state to crimped state (Supplementary Fig. 18a), and proper folding can also increase the current (Supplementary Fig. 18c). Hence, the wf-layer (18 × 6 cm, crimped state) exerts the maximum output power density of 8.86 W m$^{-3}$ and maintain stable electric power with external loads (Fig. 3h, Supplementary Fig. 19).

### Electricity generation performance of md-layer

Water escaped from wf-layer will be then absorbed by the top md-layer in mc-WEG for electricity generation at the stage II (Supplementary Fig. 5). The porous Au mesh is employed as the bottom electrode for

md-layer to receive moisture from wf-layer in mc-WEG, and the top electrode is an Au foil for electricity output (Fig. 4a). md-layers were directly placed in different humidity environments to investigate the power generation performance.

As for md-layer, the asymmetric PVA-LiCl($c_h$)/H-PSS/PVA-LiCl($c_l$) structure was designed from the bottom to top ($c_h$ denotes the concentration of LiCl in high concentration side, and $c_l$ denotes that in low concentration side). As the moisture first absorbed at bottom side, dissociated Li$^+$ and Cl$^-$ in PVA-LiCl ($c_h$) (stage I in Fig. 4b), as well as H$^+$ and the immobilized negatively-charged polyanion segments in H-PSS (stage II in Fig. 4b) will be formed. Because the ions concentration is lower at top side than the bottom of md-layer before water arriving, dissociated ions have the gradient distribution from the bottom (high concentration) to the top (low concentration) in md-layer. Therefore,

the positively-charged Li$^+$ and H$^+$ will diffuse from bottom into top under concentration difference[22,23], while the negatively-charged polyanion segments in H-PSS membrane prevent the Cl$^-$ diffusion because of the electrostatic repulsion (Supplementary Fig. 20, stage III in Fig. 4b). Finally, the positive and negative charges are separated and electricity generated (Supplementary Note 5).

The moisture induced cation (Li$^+$, H$^+$) directional migration in md-layer could be confirmed by the diode characteristics in current-voltage curve (Fig. 4c). Net ion migration is affected by both intrinsic ions flow and applied electric field. The synergy between the two fields will improves the current, otherwise the opposite, resulting in diode characteristics (Supplementary Fig. 21)[13,36]. When applying 1.8 V bias on md-layer from bottom to top, the accumulated charge in outer circuit ($Q_{bt}$) is about 1.12 mC, while only about 0.003 mC ($Q_{tb}$) when the applying bias is 1.8 V from top to bottom (Supplementary Fig. 22). The maximum rectification ratio ($Q_{bt}/Q_{tb}$) is about 382.25, exhibiting significant ion rectification in md-layer (Fig. 4d). This result illustrates that the electric field from bottom to top will induce more net ion migration in md-layer, further explicating the direction of inner cation migration is from the bottom to the top side[13,37]. It should be noted that the maximum rectification ratio of the md-layer without H-PSS membrane is only 14.87 (Fig. 4d), which indicates the negatively-charged polyanion segments in H-PSS membrane would prevent the Cl$^-$ diffusion and promote the directional migration of cation (Li$^+$, H$^+$). Meanwhile, the rectification ratio of single H-PSS in the moisture absorbing process is 220.63 that is lower than that of asymmetric PVA-LiCl(0.5 M)/H-PSS/PVA-LiCl(0.01 M) structure (Fig. 4d), which suggests the asymmetrical PVA-LiCl(0.5 M) and PVA-LiCl(0.01 M) structure will provide the Li$^+$ concentration difference from bottom to top, thus inducing enhanced cation migration. The electricity performance of md-layer with different stacking mode also indicates that the asymmetrical design will provide an enhanced potential for improving the electricity generation ability of md-layer (Supplementary Fig. 23). Above results coherently confirm our asymmetric PVA-LiCl(0.5 M)/H-PSS/PVA-LiCl(0.01 M) structure greatly benefit the moisture enabled ions directional migration in md-layer for electricity generation by pre-introduced Li$^+$ concentration and the cation selectivity of H-PSS membrane.

As expected, md-layer (1 × 1 cm) delivers $V_{OC}$ of ~0.60 V and $I_{SC}$ of ~76 μA when absorbing water through the porous electrode from 90% RH environment (Fig. 4e). With humidity increment from 31% RH to 90% RH, $I_{SC}$ increases from 12.08 to 76.32 μA (Fig. 4f), which attributes to the enhanced water absorption of md-layer under high humidity (Supplementary Fig. 24) that will generate more dissociated ions. Owing to the effect of md-layer thickness on ions diffusion[12,38], 60 μm PVA-LiCl(0.5 M) membrane, 540 μm H-PSS membrane and 40 μm PVA-LiCl(0.01 M) membrane are employed for the md-layer for further study (Supplementary Fig. 25). With the increase in the area of md-layer from 0.25 cm$^2$ to 6 cm$^2$, $V_{OC}$ is almost constant at ~0.60 V while $I_{SC}$ enhances from 20.94 μA to 105.19 μA because larger area could absorb more moisture and induce the formation of more mobile ions (Fig. 4g). The current density declines with the area increment that might be induced by the inconsistent water absorption of each region in larger area (Supplementary Fig. 26). A maximum output power density of 0.59 μW cm$^{-2}$ at an optimal resistance of 5.6 kΩ (Fig. 4h) can be exerted by md-layer (0.25 × 0.25 cm). Meanwhile, md-layer maintains ~0.60 V in $V_{OC}$ after 20 cycles of moisture absorption and desorption without significant attenuation (Supplementary Fig. 27). The output performance is stable at ~0.60 V in $V_{OC}$ as well as ~70 μA in $I_{SC}$ at 45° and 90° bending state (Supplementary Fig. 28), exhibiting better mechanical flexibility. Besides, several md-layer units can be integrated to improve the output performance, which manifests nearly linear performance increment (Supplementary Fig. 29), bringing convenience for following multistage coupling device construction and applications.

## Electricity generation, customized design and applications of mc-WEG

By integrating wf-layer, PAN membrane and md-layer together, mc-WEG can employ the multistage electricity generation by water flow and moisture diffusion synchronously in one device (Fig. 5a). When wf-layer (3 × 6 cm) and md-layer (3 × 2 cm) were integrated into mc-WEG$_1$, 0.67 V and 104.89 μA were generated by inner md-layer (3 × 2 cm) (Supplementary Fig. 30), as well as 0.24 V and 42.70 μA were generated by wf-layer (3 × 6 cm) (Supplementary Fig. 31), which are similar with the individual layers before integration. This reveals that water can be well-employed for multistage electricity generation by wf-layer and md-layer in the integrated mc-WEG$_1$, and the calculated maximum power output is ~92 mW m$^{-2}$ (or ~11 W m$^{-3}$), outperforming than other WEG (Supplementary Table 1, Supplementary Fig. 32).

In consideration of the highly adjustment of size as well as spatial form of each functional layer, it is fascinating that wf-layer and md-layer can be integrated into mc-WEG in the form of "flexible building block" (Fig. 5a) through size control, space optimization, as well as integration design. Ulteriorly, the electrical output can be customized by connection design of the inner function layer (in series or parallel) through the design of external circuits (Supplementary Fig. 33–36, Supplementary Note 10). For example, the output of mc-WEG$_1$ can be adjusted by the connection design of inner wf-layer and md-layer, which could produce 0.23 V and 202.07 μA in parallel design, as well as 0.91 V and 50.07 μA in series design (Fig. 5b) in 90% RH environment. Meanwhile, mc-WEG$_1$ reaches the maximum power density of 320.94 mW m$^{-3}$ at 3.9 kΩ external load in parallel design, while 133.08 mW m$^{-3}$ at 20 kΩ external load in series design (Fig. 5c), which signifies that mc-WEG can achieve controllable output through dual design of internal building block size and connection.

Similarly, two md-layers (0.5 × 2 cm) and one wf-layer (1 × 6 cm) can be integrated in series into the mc-WEG$_2$, which delivers 1.42 V in $V_{OC}$ and 23.47 μA in $I_{SC}$ (Supplementary Fig. 35). Meanwhile, the performance of mc-WEG$_2$ only exhibits slightly fluctuation (1.18 V − 1.93 V and 7.18 μA − 25.50 μA) under the fluctuation environment of ~33% − 84% RH and -18 °C − 34 °C (Fig. 5d) in the outdoor test in Beijing for about 3 days, embracing good environmental applicability. Besides, the electrical power is capable of charging 47 mF commercial capacitor to ~2 V by connecting three mc-WEG$_3$ (three 0.5 × 2 cm md-layers and one 1.5 × 6 cm wf-layer connected in series) in parallel (Fig. 5e). Furthermore, twenty-four mc-WEG$_3$ units connected in series with 2 meters length (Fig. 5f) could be folded into 8 × 6 × 4 cm$^3$ in volume based on the flexibility of inner layer (Fig. 5g). The mc-WEG$_3$ pack could be then directedly placed into a designed auto-switchable adsorption and desorption generating setup (Fig. 5g, Supplementary Fig. 37). The generated $V_{OC}$ is up to 36.8 V and the $I_{SC}$ can maintain at ~12 μA (Supplementary Fig. 38a, b), which can directly motivate an intelligent atomized glass (13 × 6 cm) to work normally, switching between atomized state and transparent state (Supplementary Fig. 38c).

Ulteriorly, the folding of wf-layer (3 × 6 cm) as mentioned above will enhance the output current (Supplementary Fig. 18) and further reduce the volume of device to achieve preferable performance in a restricted area. Therefore, the mc-WEG$_4$ (three 6 × 6 cm wf-layers folded five times into 1 × 6 cm and one 3 × 2 cm md-layer connected in parallel) was designed, and electrical output of 10.32 V (Fig. 5h) and ~280 μA (Supplementary Fig. 39) can be achieved when twenty-two mc-WEG$_4$ units connected in series, which can actuate a series of current-driven devices like table lamp and LED strip (six LEDs in parallel) continuously working more than 30 min respectively without charging the capacitor in advance (Fig. 5i, Supplementary Movie 1), being discriminated from previous MEGs[12,15,39,40], which can only induce the electrical appliance to work for a short time after charging the capacitor for a long time. More complex mc-WEG designs and output performance can be acquired in Supplementary information, which

exhibits favorable power generation ability, exceeding the performance of other hydropower devices at the optimum external load (Supplementary Table 2, Supplementary Fig. 40)[12,13,15,19,22–24,33,34,37,39,41–43].

## Discussion

In summary, the multistage coupling water-enabled electric generator has been successfully developed by employing the internally liquid flow and moisture diffusion synchronously for electricity generation. Based on the well combination of multistage functional layers and distributary transport for internal water in different forms, mc-WEG delivers a maximum output power density of ~92 mW m$^{-2}$ (~11 W m$^{-3}$). A three-stage adjustment strategy of size control, space optimization, and integration design has further endowed the high degree of freedom for various mc-WEGs construction to meet the diverse requirements for different electric applications. Twenty-two well-regulated mc-WEG units connected in series can achieve ~10.32 V and ~280 µA output, directly driving a table lamp to continue working more than 30 min without charging the capacitor in advance. mc-WEG also embraces fascinating flexibility and environmental adaptability, which can maintain stable electrical output after being folded and exposed in natural environments. This work suggests that multistage energy in water circulation can be utilized through reasonable WEG design, which will deliver promising development of green water energy.

## Methods

### Materials

Cotton fabric (~200 µm thickness), Ketjen black powder (Lion Corporation, ECP), cetrimounium bromide (CTAB, Adamas, 99%), Polystyrene sulfonic acid (PSS, 30 wt% in water, $M_w$ ~ 75,000, Energy Chemical), polyvinyl alcohol (PVA-1788, alcoholysis degree 87.0 ~ 89.0), LiCl (Adamas, 99%), fluorescein sodium (Energy Chemical), polyacrylonitrile (PAN, $M_w$ 150,000), N,N-Dimethylformamide (DMF, Innochem, ≥99.5%GC), Polyimide tape (Dongguan Xinshi Packaging Materials Co., LTD).

### Preparation of wf-layer

0.2 g of Ketjen black powder and 0.4 g CTAB were dispersed in 40 ml deionized water to form the carbon ink. Cotton fabric was cut into different size and infiltrated into carbon ink for various times to form the carbon fabric with different resistance. Then, part of the carbon fabric was immersed in CaCl$_2$ aqueous solution (25 wt%) and dried at 80 °C to obtain wf-layer. The two ends of wf-layer are connected with Au electrodes for electricity output.

### Preparation of PAN membrane

For the preparation of PAN solution, PAN (7.5 g) was dispersed into DMF (42.5 g) with vigorous stirring using magnetic stirrers at 90 °C for 24 h. During electrospinning, the as-prepared solution was added into plastic syringe and a high direct current voltage of 20 kV was applied to needle tip. The as-spun fibrous membranes were collected on the grounded metallic rotating roller covered by an Au mesh. The distance between the collector and the needle tips was 20 cm.

### Preparation of md-layer

Aqueous PSS dispersion (15 wt%) was uniformly mixed with 0.05 M H$_2$SO$_4$ with the same volume. The mixed solution was cast into petri dish and dried in an oven at 45 °C 60% RH. LiCl with certain amount was uniformly mixed with 10 wt% PVA. The concentration of LiCl is calculated based on the PVA solution before drying. The mixed solution was cast into petri dish and dried in an oven at 80 °C 20% RH. During the test, the obtained md-layer were sandwiched between a pair of asymmetry Au electrodes (Au mesh and Au foil).

### Fabrication of mc-WEGs

The mc-WEG was composed with wf-layer, PAN membrane on porous Au electrodes, and md-layer membrane in sequence. Part of the mc-WEG were packed by Polyimide tape.

### Measurement

The permeability test was investigated in accordance with GB/T 1037-2021. The water absorption test was conducted in compliance with GB/T 21655.1-2008. So as to maintain the good spread of PAN membrane, the saturated water absorption of PAN membrane with a substrate (Au mesh) was first researched, and then the saturated water absorption of Au mesh was studied. The saturated water absorption of PAN membrane can be obtained by difference analysis. The electrical output measurements were performed with Keithley 2400 and Keithley 2612b (Keithley Instruments). The circuit parameter of current during the voltage output test was 0 nA, and the circuit parameter of voltage during the current output test was 0 V.

## Data availability

The data generated in this study are provided in the Supplementary Information and Source Data file. Source Data are provided with this paper.

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

## Acknowledgements

This work was supported by the financial support from the National Natural Science Foundation of China No. 52022051, 22035005, 22075165, 52090032 and 52073159 (52022051, 22075165 and 52090032 to H.C., 22035005 and 52073159 to L.Q.), Tsinghua-Foshan Innovation Special Fund (2020THFS0501, to L.Q.).

## Author contributions

L.Q., H.C. and P.L. designed the experiments and accomplished the original draft. Y.H., W.H., B.L. and H.W. gave advice on experiments. L.Q. and H.C. supervised the entire project. All authors discussed the results and reviewed the manuscript.

## Competing interests

The authors declare no competing interests.
