## [Peer Review File · Nature Communications]

Multistage coupling water-enabled electric generator with customizable energy outputREVIEWER COMMENTS

Reviewer #1 (Remarks to the Author):

This work designed a novel water-enabled electric generator, and the multistage coupling design for energy harvesting is very attractive. As far as I know, previous hydrovoltaic device and water-enabled electric generator can only use single process in water circulate like water liquid water evaporation, moisture diffusion, or water flow for electricity generation. Mc-WEG in this manuscript realized the simultaneous liquid and gaseous water transmission for cooperative energy output in one device. The rationally-constructed water distributary made the water flow induced electricity generation at bottom layer and moisture induced electricity generation at top layer, achieving the stereoscopic water resource utilization and outstanding energy performance. The "flexible building block" strategy is also very interesting, which will provide customized mc-WEG construction and electricity output beyond similar reported devices. This manuscript is meticulously organized, and results are consistent with the discussion well. The application design and display are so exciting. Therefore, this work is recommended for publication highly, which will be of great interest for hydrovoltaic technology and water-enabled electricity generation in future.

Reviewer #2 (Remarks to the Author):

The authors reported the design of a water-enabled electric generator by coupling a water flow-induced electricity generator with a moisture diffusion-induced electricity generator. The idea is novel and there are plenty of data. However, scientific advancement is inferior. Part of the data is debating. The experimental description is not sufficient to understand the work. There are language issues. Overall, the current manuscript does not meet the quality of Nature Communications. Part of the comments are listed below.

- 1) The Multistage Coupling Water-enabled Electric Generator with Au and carbon cloth as the current collector, what is the contribution of current collects to the device voltage generation? For the md-layer, what is the contribution from the PVA-LiCl layer by moisture adsorption from the environment directly?
- 2) How are the voltage and current collected from CaCl₂-carbon cloth, from the ends of CaCl₂ containing and non-CaCl₂ along the cloth? If yes, then what is the contribution of CaCl₂ to the voltage due to the potential difference between CaCl₂ and carbon? In the mc-WEG device, voltage and current collection seem to be perpendicular to cloth thickness according to the schematic. Then the question is how to correlate the description of a separate wf device with the wf-layer in the mc-WEG? There is a discussion about the loading of CaCl₂ on device performance, what is the influence of CaCl₂ length? What is the electronic conductivity and ionic conductivity of the carbon cloth and CaCl₂-coated carbon cloth? How much does the pure cotton fabric contribute to the voltage and current generation?
- 3) Why Figure 2e shows 2 images of 1h sample? In 2h, what does scale 3.0 m mean?

Reviewer #1 (Remarks to the Author):

This work designed a novel water-enabled electric generator, and the multistage coupling design for energy harvesting is very attractive. As far as I know, previous hydrovoltaic device and water-enabled electric generator can only use single process in water circulate like water liquid water evaporation, moisture diffusion, or water flow for electricity generation. mc-WEG in this manuscript realized the simultaneous liquid and gaseous water transmission for cooperative energy output in one device. The rationally-constructed water distributary made the water flow induced electricity generation at bottom layer and moisture induced electricity generation at top layer, achieving the stereoscopic water resource utilization and outstanding energy performance. The “flexible building block” strategy is also very interesting, which will provide customized mc-WEG construction and electricity output beyond similar reported devices. This manuscript is meticulously organized, and results are consistent with the discussion well. The application design and display are so exciting. Therefore, this work is recommended for publication highly, which will be of great interest for hydrovoltaic technology and water-enabled electricity generation in future.

Reply: We greatly thank the referee for positive recommendation for our manuscript.

Reviewer #2 (Remarks to the Author):

The authors reported the design of a water-enabled electric generator by coupling a water flow-induced electricity generator with a moisture diffusion-induced electricity generator. The idea is novel and there are plenty of data. However, scientific advancement is inferior. Part of the data is debating. The experimental description is not sufficient to understand the work. There are language issues. Overall, the current manuscript does not meet the quality of Nature Communications. Part of the comments are listed below.

Reply: We thank the referee for the positive comments and constructive suggestions, and we would like to supplement the highlights of this study here before responses:

First, **the developed multistage coupling water-enabled electric generator (mc-WEG) embraces novel multi-stage working principle.** Compared with previous water-enabled electric generator by utilizing a single interaction between functional materials and different forms' water, mc-WEG firstly achieve the utilization of two forms of water in a single device through the internal structure design for the distributary of liquid and moisture.

Second, **mc-WEGs exhibit outstanding and customizable output performance.** Because of the good flexibility and collaboration of the two functional layers, mc-WEGs can maintain good performance under close packing conditions. Therefore, a maximum output power density approaches to $\sim 91.77 \text{ mW m}^{-2}$ ($\sim 10.92 \text{ W m}^{-3}$) can be achieved, which is also leading among recently reported water induced power generators using inert electrodes (Table R1).

Table R1. Output power density of water induced power generators with inert electrodes.

No.	Material	Form of water	P_{output} (mW m ⁻²)	Ref
1	Graphene oxide membrane	Moisture	0.018	Energy Environ. Sci. 11 , 2839-2845 (2018)
2	Graphene oxide and sodium polyacrylate	Moisture	0.07	Energy Environ. Sci. 12 , 1848-1856 (2019)
3	PSS/PVA film	Moisture	7.90	Nano Energy 67 , 104238 (2020)
4	Protein nanowires	Moisture	0.0405	Nature 578 , 550-554 (2020)
5	Sodium alginate /SiO ₂ /GO	Moisture	12.00	Adv. Mater. 34 , 2106410 (2022)
6	Asymmetric hygroscopic of carbon fabric	Moisture	~70	Nat. Commun. 13 , 3643 (2022)
7	Ionic polymer Nafion and poly(N-isopropylacrylamide) hydrogel	Moisture	10.14	Energy Environ. Sci. 15 , 2489-2498 (2022)
8	Carbon	Liquid	0.053	Nat. Nanotechnol. 12 , 317-321 (2017)
9	Al ₂ O ₃ nanoparticles	Liquid	0.513	ACS Appl. Mater. Inter. 11 , 30927-30935 (2019)
10	AAO and ionic liquid	Liquid	12.1	Adv. Funct. Mater. 32 , 2203666 (2022)
11	Carbon nanoparticle	Liquid (H ₂ O)	0.26	ACS Nano 13 , 12703-12709 (2019)
		Liquid (CaCl ₂)	0.505	Energy Environ. Sci. 13 , 527-534 (2020)
12	mc-WEG	Moisture and liquid	91.77	This work

Third, inner wf-layer and md-layer can be assembled as "flexible building blocks" for the construction of specially appointed mc-WEGs through a three-stage adjustment strategy of size control, space optimization, and large-scale integration. As a result, the output electricity of mc-WEGs can be customized by rational connection design of the inner function layer and the further integration of mc-WEG units to satisfy the requirements of different electrical appliances. For example, twenty-two well-regulated mc-WEG units connected in series can achieve ~10.32 V and ~280 μ A output, directly driving a table lamp to continue working more than 30 minutes

without charging the capacitor in advance. The designed mc-WEG unit also has a leading position in the current hydro power generation devices (Figure R2).

Figure R1. Comparison of output volumetric power density and internal resistance between mc-WEG in this work and reported water induced generators. The numbers correspond to reference numbers in Supplementary Table 2 in Supplementary Information.

Therefore, our study is of significant value for promoting the development of materials science and device design in the booming water-enabled electric generators. Indeed, there are plenty of data in the main text and supporting information, and as suggested, we have improved the description and diagrammatic sketch in the revised manuscript, which would help readers to understand our work easily. We hope and believe that our paper is meaningful and appealing to the broad readership of *Nature Communications*.

1) The Multistage Coupling Water-enabled Electric Generator with Au and carbon cloth as the current collector, what is the contribution of current collects to the device voltage generation?

Reply: Thanks for the constructive comment. To demonstrate the electrodes situation in mc-WEG, we have improved the schematic diagram of mc-WEG. As shown in Figure R2, mc-WEG is composed of wf-layer for liquid flow enabled electricity generation, the hydrophilic polyacrylonitrile (PAN) membrane as the diversion layer

for moisture permeation from wf-layer to md-layer, and md-layer for moisture enabled electricity generation. For wf-layer, Au electrodes are on the left and right ends of wf-layer in the schematic diagram. Carbon cloth is one part of the wf-layer and does not act as the current collector. For md-layer, a porous Au mesh is employed as the bottom electrode to receive moisture from wf-layer in mc-WEG, and the top electrode is an Au foil. Therefore, the electrodes used for the electricity output in mc-WEG are all Au electrodes, and the output can be collected by connecting the electrodes of md-layer and wf-layer in series or parallel through the design of external circuits. Au electrodes, as current collectors, are chosen because of their good conductivity and chemical stability, which will not contribute to the voltage generation.

Peer your suggestions, we have revised and highlight the structure diagram of mc-WEG in Supplementary Fig. 4 and related description in the main text.

Figure R2. Structure diagram of mc-WEG, which is composed of wf-layer, PAN membrane, and md-layer corresponding with Au electrodes configuration.

For the md-layer, what is the contribution from the PVA-LiCl layer by moisture adsorption from the environment directly?

Reply: When integrating wf-layer, PAN membrane and md-layer together in mc-WEG (Figure R1), only the CaCl_2 loaded region of wf-layer will be exposed in environment, and the rest of the parts are packaged with Polyimide tape. Therefore, the PVA-LiCl layer will not directly absorb moisture from environment in mc-WEG.

2) How are the voltage and current collected from CaCl₂-carbon cloth, from the ends of CaCl₂ containing and non-CaCl₂ along the cloth? If yes, then what is the contribution of CaCl₂ to the voltage due to the potential difference between CaCl₂ and carbon? In the mc-WEG device, voltage and current collection seem to be perpendicular to cloth thickness according to the schematic. Then the question is how to correlate the description of a separate wf device with the wf-layer in the mc-WEG?

Reply: As indicated in Figure R2, Au electrodes are connected at the ends of wf-layer (CaCl₂ loaded region and unloaded region). The voltage and current of wf-layer in mc-WEG are collected through Au electrodes as same as that in the separate wf-layer. The output of the whole mc-WEG can be collected by connecting the electrodes of inner md-layer and wf-layer in series or parallel through the design of external circuits (Figure R3) for different demands. Meanwhile, to evaluate the potential difference between CaCl₂ loaded region and unloaded region, the voltage (Figure R4a) and current (Figure R4b) of wf-layer under dry condition (RH = 15%) are measured, exhibiting no electrical signals, which could not contribute to the voltage generation.

Figure R3. The connection mode between md-layer and wf-layer of mc-WEG when connected in series (a) and in parallel (b).

There is a discussion about the loading of CaCl₂ on device performance, what is the influence of CaCl₂ length? What is the electronic conductivity and ionic conductivity of the carbon cloth and CaCl₂-coated carbon cloth? How much does the pure cotton fabric contribute to the voltage and current generation?

Reply: First, additional experimental supplements of the influence of CaCl₂ loaded length on the electrical performance have been carried out. As can be seen from Figure R5, the voltage and current increased from 0.12 V to 0.49 V and 16.89 μA to 66.22 μA as the CaCl₂ loaded length increased from 1 cm to 4 cm in the wf-layer unit (3 cm×6 cm), because more water could be absorbed by wf-layer with longer CaCl₂ loaded length.

Figure R5. Electricity performance of wf-layer (3 cm×6 cm) with different CaCl₂ loaded length.

Second, the conductivity of the carbon fabric and CaCl₂-coated carbon fabric are measured by assembling into a button battery form by Au electrode. As indicated in the I-V curves (Figure R6a), the resistance is calculated by the formula ($R=U/I$) and the conductivity σ can be obtained ($\sigma=d/(R \times A)$, where d is the thickness of the carbon fabric, and A is the facing area between electrodes) (Figure R6b). The calculated conductivity of carbon fabric and CaCl₂-loaded carbon fabric is about 0.037 S cm⁻¹ and 0.049 S cm⁻¹, respectively. Because of the existence of functional groups (-COOH, -OH), which will dissociate after absorbing water, the conductivity of wet carbon fabric will increase to about 0.10 S cm⁻¹. As for CaCl₂-loaded carbon fabric, large amounts of Ca²⁺ and Cl⁻ will be dissociated after absorbing water, so the conductivity will increase to 0.14 S cm⁻¹.

Figure R6. The current-voltage curves (a) and related conductivities (b) of carbon fabric and CaCl₂-coated carbon fabric before and after absorbing water.

Third, we have measured the pure cotton fabric under the test condition of wf-layer (Figure R7). However, there is no voltage and current generated. Therefore, pure cotton fabric could not contribute to the voltage and current generation in our device.

Figure R7. The voltage-time curve (a) and current-time curve (b) of pure carbon fabric under the test condition of wf-layer.

3) Why Figure 2e shows 2 images of 1h sample? In 2h, what does scale 3.0 m mean?

Reply: For Fig. 2e, we have changed the picture marked with 1 h into a picture marked with 6 h. The scale in Fig. 2h represents the mass (mg) and the figure has been improved in the revised manuscript. Thank you very much for pointing out this.

REVIEWERS' COMMENTS

Reviewer #1 (Remarks to the Author):

The authors have revised and improved the manuscript according to the suggestions, and the manuscript is recommended for publication.

Reviewer #2 (Remarks to the Author):

The authors addressed my concerns. In addition, I'm curious about the insight from the authors on the contribution of water phase change in the device to the performance.

Reviewer #1 (Remarks to the Author):

The authors have revised and improved the manuscript according to the suggestions, and the manuscript is recommended for publication.

Reply: We really appreciate the referee for the comments.

Reviewer #2 (Remarks to the Author):

The authors addressed my concerns. In addition, I'm curious about the insight from the authors on the contribution of water phase change in the device to the performance.

Reply: We really appreciate the referee for the comments.